# Preliminary In Vitro Evaluation of Chitosan–Graphene Oxide Scaffolds on Osteoblastic Adhesion, Proliferation, and Early Differentiation

**DOI:** 10.3390/ijms21155202

**Published:** 2020-07-22

**Authors:** Sonia How Ming Wong, Siew Shee Lim, Timm Joyce Tiong, Pau Loke Show, Hayyiratul Fatimah Mohd Zaid, Hwei-San Loh

**Affiliations:** 1School of Biosciences, Faculty of Science and Engineering, University of Nottingham Malaysia, Jalan Broga, Semenyih Selangor Darul Ehsan 43500, Malaysia; son1a.93.sw@gmail.com; 2Department of Chemical and Environmental Engineering, Faculty of Science and Engineering, University of Nottingham Malaysia, Jalan Broga, Semenyih Selangor Darul Ehsan 43500, Malaysia; joyce.tiong@nottingham.edu.my (T.J.T.); pauloke.show@nottingham.edu.my (P.L.S.); 3Fundamental and Applied Sciences Department, Centre of Innovative Nanostructures & Nanodevices (COINN), Institute of Autonomous System, Universiti Teknologi PETRONAS, Bandar Seri Iskandar 32610, Malaysia; hayyiratul.mzaid@utp.edu.my

**Keywords:** graphene oxide, chitosan, western blot, alkaline phosphatase

## Abstract

An ideal scaffold should be biocompatible, having appropriate microstructure, excellent mechanical strength yet degrades. Chitosan exhibits most of these exceptional properties, but it is always associated with sub-optimal cytocompatibility. This study aimed to incorporate graphene oxide at wt % of 0, 2, 4, and 6 into chitosan matrix via direct blending of chitosan solution and graphene oxide, freezing, and freeze drying. Cell fixation, 3-(4,5-dimethylthiazol-2-yl)-2,5-diphenyl tetrazolium bromide, alkaline phosphatase colorimetric assays were conducted to assess cell adhesion, proliferation, and early differentiation of MG63 on chitosan–graphene oxide scaffolds respectively. The presence of alkaline phosphatase, an early osteoblast differentiation marker, was further detected in chitosan–graphene oxide scaffolds using western blot. These results strongly supported that chitosan scaffolds loaded with graphene oxide at 2 wt % mediated cell adhesion, proliferation, and early differentiation due to the presence of oxygen-containing functional groups of graphene oxide. Therefore, chitosan scaffolds loaded with graphene oxide at 2 wt % showed the potential to be developed into functional bone scaffolds.

## 1. Introduction

Tissue engineering was a novel solution to the misery underlying the limitations of the autografts and allografts approach in regenerating bone defect areas, either arising due to congenital anomalies or induced traumatically [1]. Among interesting biological observations leading to the concept of tissue engineering is the in vitro genesis of tubular network that closely resembles the capillary vasculature in vivo from capillary endothelial cells under appropriate culture conditions [2]. The three main elements which constitute a tissue engineering construct are cells, biomaterials, and specific growth signals that dictate cell proliferation and differentiation into a specialized tissue [3]. 

Biomaterials serve as biomimetic extracellular matrices (ECM) [4] which function as temporary scaffold for tissue development and regulate tissue-specific gene expression by releasing growth factors to the adjacent cells [5]. An ideal scaffold as a transient ECM generally requires good mechanical properties, biocompatibility, microstructure, tailorable degradation rates [6], and relatively nontoxic degradation products [7]. Among biomaterials, scaffolds made of natural polymers are highly sought-after.

Chitosan (CS), a *N*-deacetylated chitin derivative, is a promising material for bone tissue engineering owing to its structure resemblance to the glycosaminoglycans (GAGs) found in natural bones and cartilage [8,9]. It has been characterized with exceptional biocompatibility, biodegradability, non-toxicity [10], as well as antimicrobial activities [11,12]. The cationic nature of multiple amino groups in CS facilitated the functionalization with anionic GAGs which in turn concentrated cytokines or growth factors [13]. Besides, its superiority over synthetic polymers offers a platform for new tissue growth which conquers its load-bearing functions [14]. Notably, CS/fibroin-hydroxyapatite membrane promoted guided bone regeneration in rat which further verified the in vivo efficacy of CS [15]. Wang and Stegemann discovered the correlation between the osteogenic differentiation of human bone marrow mesenchymal stem cells (hBMSC) and CS content in CS-collagen composites based on alkaline phosphatase activity, an early osteoblast differentiation marker [16]. Nonetheless, they also underscored CS’s inferior cytocompatibility, possibly due to the nonexistence of cell attachment binding sites. Despite the hopeful qualities offered by CS, one of its major drawbacks includes its non-supportive nature in cell adhesion which is a key factor in osteoconductivity for bone regeneration [17]. 

Recently, graphene has emerged as vital material from electronic to biomedical field. As a single layer of sp^2^ bonded carbon atoms arranged in a two-dimensional hexagonal matrix, graphene has huge surface area and excellent mechanical properties [18,19,20]. Moreover, its derivative graphene oxide (GO) produced from random introduction of carboxyl, epoxide, and hydroxyl groups have even improved its hydrophilicity, aqueous dispersibility, and colloidal stability [21]. All these properties have extended its use in the biomedical field. 

This study aimed to reinforce CS scaffolds with GO at wt % of 2, 4, and 6 which were further crosslinked with glutaraldehyde (GTA). The highest GO wt % attempted in this study was justified by a recent work by Aidun and coworkers (2019) [22]. They also prepared their chitosan composite scaffolds with GO at wt % of 6. Further GTA crosslinking on CS-GO scaffolds would be expected to show structural integrity to facilitate osteoblastic functions like adhesion, proliferation and early differentiation. However, the preliminary in vitro efficacy of GTA crosslinked CS-GO scaffolds has not been proven which would be verified in this study. The in vitro efficacy of these nanocomposite scaffolds was evaluated with human osteosarcoma cell line MG63, a classic model cell line for osteoblast functional studies. This cell line retains the vital markers of osteogenic cell differentiation like alkaline phosphatase (ALP) [23,24].

## 2. Results

### 2.1. Characterization of GO

Figure 1a shows the two-dimensional geometry of GO. The surface of GO was observed with wrinkles and folds. The main elements of GO (in purple box) detected by EDX included carbon, oxygen, and trace amount of sulfur which were tabulated in Table 1. The UV–VIS absorption spectrum of GO suspended in deionized water is shown in Figure 1b. The main peak of GO was observed at wavelength of 230 nm confirming π → π* transition of aromatic C-C bond [25].

### 2.2. Morphology Study of GTA Crosslinked CS-GO Scaffolds

The pristine GTA crosslinked CS scaffold in Figure 2a reveals a rather smooth surface. However, those with the GO dispersion, CS-GO2, CS-GO4, and CS-GO6 (Figure 2b–d), exhibited certain degree of surface roughness. Another noteworthy point these images reveal the tangible collapse of the interconnected pores which should have been formed through the water crystal formation and withdrawal via freezing and freeze-drying, respectively.

### 2.3. FESEM Images of MG63 Cells Fixed on Scaffolds for Cell Adhesion Study

Figure 3 shows MG63 cells on CS-GO2 (Figure 3b) having a wider spread over the scaffold surface, while CS, CS-GO4, and CS-GO6 (Figure 3a,c,d respectively) manifested comparatively more rounded morphology with numerous cellular filopodia suggesting the cytocompatibility on these scaffolds. 

### 2.4. MTT Assay

As presented in Figure 4, CS-GO2 gives promising cell viability percentage of 95.84% on the third day post-seeding and shows significant difference compared to CS scaffolds (* *p* < 0.001). On the other hand, as the incubation period increased to 7 and 14 days, CS scaffolds at all GO concentrations gave no significant difference (*p* > 0.05) in cell viability from MG63 cell control. 

### 2.5. ALP Assay

Figure 5 shows the ALP activity at the absorbance of 405 nm after 2 h of ALP-pNpp incubation. ALP activities of MG63 cells on all scaffolds were observed on each day post-incubation. GO loading may have enhanced the osteogenic properties of GTA crosslinked CS scaffolds in terms of ALP activities. Therefore, more studies will be required to verify this.

On as early as 7th day post-seeding, MG63 cells exhibited a significant (** *p* < 0.001) level of ALP activity on CS-GO2 and CS-GO6 in contrast to CS scaffolds. A remarkable observation could be made on the ALP activity of MG63 on CS-GO6 on 28th day post-incubation. The ALP activity started off at a high (*p* < 0.001) 0.709 µM/mL min^−1^ which then plummeted on 21st day post-incubation before ascending to the level of ALP activity similar to that achieved on 7th day post-incubation by 28th day of cell-scaffold interaction (Figure 5). The decrease in ALP activity observed in CS-GO6 scaffolds at day 21 may be caused by the presence of a greater number of mature osteoblast cells as compared to those undergoing early differentiation [26]. On day 28, the increased ALP activity of MG63 on CS-GO6 was observed again which implies bone formation. Therefore, more investigation will be required to confirm this speculation. 

### 2.6. Western Blot Detection of Alkaline Phosphatase (ALP)

Based on the nitrocellulose membrane capture from western blot assay shown in Figure 6, the early osteoblast differentiation marker, ALP, was detected at the correct band size of 80 kDa (as highlighted in red box), the tissue non-specific type, TNSALP [27]. 

## 3. Discussion

In the endeavor of bone tissue engineering (BTE), a primary criterium lies within its ability to sustain the cell penetration, proliferation, and differentiation in bone remodeling [28]. The FESEM images in Figure 2 tangibly show collapse of the interconnected porous structure which should have been formed via the freezing and freeze-drying processes. This could be caused by the low viscosity of the CS solution which contributed to the rapid freezing rate during the quench of CS-GO solution to −20 °C before water droplets could coalesce for the formation of ice crystals [29]. Besides, the wt % of GO is not high enough to retain the porous structure of CS scaffolds. CS scaffolds with TiO_2_ nanotubes as filler in a previous study showed porous structure [30]. The wt % of TiO_2_ nanotubes in the scaffolds was as high as 16 wt %. However, the wt % of GO has to be kept not more than 6 in this study to ensure the cytocompatibility of CS scaffolds. Therefore, the resultant scaffold is a stack of CS layers without visible pores. Nonetheless, all scaffolds of CS and CS-GO demonstrated certain degree of roughness on their surfaces (Figure 2). 

GO comprises a large surface area supplemented with a panoply of oxygen-containing functional groups. It is suggested that GO dispersion is more even in CS solution containing 2 wt % of GO. When the wt % of GO is increased to 4 and 6, the agglomeration of GO becomes dominant due to high surface energy. The dispersive force by magnetic stir bar is not strong enough to prevent the agglomeration. Instead of being well dispersed in the solution, GO stay as clumps and deposit at other parts of CS-GO scaffolds. The release of GO clumps in CS scaffolds starts upon the degradation of chitosan matrix. Even the clumps are not released, they remain in the chitosan matrix facilitating sites for osteoblastic adhesion which will be followed by proliferation and differentiation. However, more investigation is needed to confirm the presence of GO particles in other parts of CS scaffolds.

As depicted in Figure 3b, MG63 cells also demonstrated a conspicuous flattened morphology on CS-GO2 in contrast to a more spherical morphology on other scaffolds. The oxygen-containing functional groups such as hydroxyl (-OH) groups on CS-GO2 scaffolds are suggested to be responsible for better adaptability of MG63 to CS-GO2. These functional groups enhance the polarity and wettability of the surface [31,32], thereby allowing cell adhesion-mediating molecules such as fibronectin and vitronectin to form ionic interaction with it [33] in forms with more flexibility for the strategic positioning of various active sites. In addition, the increased wettability contributed by hydroxyl groups has shown to reduce the adsorption of albumin which is non-adhesive for cells [34]. 

When the findings from MG63 cells initial adhesion to the scaffolds are combined with MTT and ALP data, CS-GO2 was found to not just improve MG63 cell spreading and adhesion on the scaffold but also significantly (* *p* < 0.001) enhanced the cell viability percentage against CS on the third day post-seeding (Figure 4). Moreover, on as early as seventh day post-incubation in osteoinductive medium, CS-GO2 also gives a significantly high (** *p* < 0.001) ALP activity in contrast with CS (Figure 5). CS-GO6 also gave promising enhancement of ALP activity both on 7th and 28th day post-seeding (Figure 5) although the initial cell viability percentage was only 53.3% of MG63 cell control (Figure 4). This is contradictory to a recent finding which reported that the ALP expression of MG63 was decreased over incubation period with GO nanosheets [35]. Also, the administration of nanographene platelets was also shown to hamper osteogenesis from as low as above 10 µg mL^−1^ [36]. Nonetheless, these previous works were conducted using electrodeposited CS-GO film and direct GO solution administration, which could potentially bring the final GO concentration exposed to MG63 cells much higher, in contrast to this study whereby GO was incorporated into CS matrix to form a 3D structure prior to exposure to cells. Therefore, the final GO concentration exposed to MG63 cells is deduced to be lower in this study. 

The mechanism behind the enhancement of osteogenesis brought about by CS-GO6 is hypothesized to stem from the chemical properties conferred by higher GO loading. Some studies supported that GO being an osteoinductive material even in the absence of osteogenic medium [37,38]. Most of the findings have been ascribed to the various planar oxygen-containing functional groups like hydroxyl, carboxyl, and epoxide which have allowed various π–π stacking interactions, electrostatic and hydrogen bonds for the concentration of osteogenic inducers [39] and growth factors [40]. Besides, the osteoinducer used in this study, l-ascorbic acid, was also suggested to adsorb onto GO and subsequently binds calcium ions which mediated increase in ALP activity and mineralization in mouse osteoblasts [41]. 

## 4. Materials and Methods

### 4.1. Characterization of GO

The surface and elemental composition of GO were examined at accelerating voltage of 20 kV using a field emission scanning electron microscope (FESEM, FEI, Quanta 400F, Hillsboro, OR, USA) and energy dispersive X-ray (EDX, Oxford-Instrument INCA 400 with X-Max Detector, High Wycombe, ENG, UK) respectively. The UV–VIS absorption spectrum of suspended GO was conducted at range of 200–800 by using UV–VIS spectrophotometer (PerkinElmer, Lambda 25, Boston, MA, USA). GO was dispersed in deionized water for 5 min by using an ultrasonic bath (Elmasonic S180H, Singen, BW, Germany) prior to UV–VIS analysis.

### 4.2. Fabrication of GTA Crosslinked CS-GO Scaffolds

The CS-GO composite scaffolds were formulated via the direct blending, freezing and freeze-drying methods. Three wt % (wt %) of CS (Sigma-Aldrich, Saint Louis, MO, USA) solution in 0.2 M acetic acid was blended with (0, 2, 4, 6 wt %) of GO (Nalgene Graphene Supermarket, Ronkonkoma, NY, USA) via direct blending for 20 min to produce CS composite scaffolds with different wt % of GO as tabulated in Table 2. Prior to blending with CS, different wt % of GO were suspended in deionized water for 5 min using ultrasonic bath. Subsequently, the CS-GO mixture was solidified at −20 °C and freeze-dried at −40 °C for 24 h each. The CS-GO scaffolds were rehydrated with 100%, 70%, and 50% ethanol for 1 hour, 30 min, and 30 min, respectively. The rehydration of scaffolds using alcohols allowed direct sterilization of scaffolds and hydrated CS scaffolds were reported to be spongy and very flexible [42]. The CS-GO scaffolds were dried with silica gels for 2–3 days. The desiccated scaffolds were then crosslinked by immersing them in 100 mL 1% (*v*/*v*) GTA solution for 20 h. Each of the crosslinked scaffolds was then rinsed with three washes of 200–300 mL of deionized water and subsequently desiccated with silica gels until complete dryness. A final step of sterilizing each crosslinked scaffold in 1 mL of 70% (*v*/*v*) ethanol for 30 min was conducted prior to characterization assays on their in vitro biological properties. 

### 4.3. Field Emission Scanning Electron Microscopy (FESEM) and Energy Dispersive X-Ray Spectrometry (EDX) Examination on CS and CS-GO Scaffolds

The surface morphology and elemental composition of GTA crosslinked CS-GO scaffolds was studied by using FESEM and EDX. The analyses were conducted at voltage of 5–10 kV under low vacuum condition.

### 4.4. Cell Seeding on GTA Crosslinked CS and CS-GO Scaffolds

MG63 cells were thawed from a frozen stock and cultured in 75 cm^3^ flasks using Minimum Essential Medium (MEM, Gibco, Carlsbad, CA, USA) supplemented with 10% heat-inactivated fetal bovine serum (FBS, Gibco, Carlsbad, CA, USA), 1% penicillin-streptomycin, 1% l-glutamine, and 1 mM sodium pyruvate (Sigma-Aldrich, St Louis, MO, USA) at 37 °C in a humidified atmosphere of 5% CO_2_. Cells were passaged at 80% confluency using 0.25% (*w*/*v*) trypsin (Sigma-Aldrich, St Louis, MO, USA). Fresh medium was replenished at every 2–3 days. Prior to cell seeding onto scaffold at cell density of 30,000 cells per well/scaffold, each scaffold was immersed in 1 mL MEM for 30 min for swelling. The cell-scaffold structures were maintained in MEM with 10% FBS and incubated under standard culture conditions for each duration per incubation (d.p.i.). MEM with 10% FBS supplemented with β-glycerolphosphate (Sigma-Aldrich, St Louis, MO, USA) and L-ascorbic acid (Fluka, Germany) was used in alkaline phosphatase (ALP) assay. Replicate wells containing only cell suspension served as controls. 

### 4.5. Cell Adhesion Assay

After 6 h of cell seeding onto scaffolds, the cell-scaffold construct was rinsed with 1× phosphate buffered saline (PBS, Sigma-Aldrich, St Louis, MO, USA) once. Then, the fixing solution was added to fully immerse each scaffold and incubated at 4 °C for 30 min. One PBS wash was conducted again prior to the dehydration at room temperature using ethanol of graded concentrations 70%, 95%, and 99.5% for 20 min, respectively. Subsequently, the final two drying steps were executed by immersing scaffolds in the solution of 50% pure ethanol mixed with 50% hexamethyldisilazane (HMDS, Sigma-Aldrich, St Louis, MO, USA) and 100% HMDS and gentle continuous rocking for 5 and 10 min, respectively. Excess HMDS was removed and scaffolds were air dried for approximately 2 days. The morphology of cells grown on scaffolds with different wt % of GO was examined by using FESEM at accelerating voltage of 10 kV.

### 4.6. MTT Assay

After the incubation period of 3, 7, and 14 days, respectively, the medium was replaced with MEM without FBS, 100 µL of 5 mg mL^−1^ of sterile MTT solution in PBS was added to each well and allowed to react with the cells on scaffold for 4 h. Mitochondrial dehydrogenase in viable cells cleaved MTT to form purple formazan crystals [43] and these were solubilized with DMSO added at 1 mL to each well. The plate was oscillated for maximal dissolution effect of the DMSO. The optical density at 570 nm of the resultant solution was measured using Varioskan^TM^ multiplate reader (ThermoFisher Scientific, Carlsbad, CA, USA). Quantity of formazan is presumably directly related to the number of viable cells [44]

The result was presented as cell viability percentage (%) against MG63 control cultured on polystyrene well for each incubation period.

### 4.7. ALP Assay

This colorimetric test is based on the color visibility of a reaction catalyzed by the enzyme, ALP which is an early osteoblast differentiation marker. First, both the cells and cell-scaffold construct were washed with 1× PBS before being homogenized in 1 mL cell lysis buffer (PRO-PREP^TM^, Gyeonggi-do, Korea) and incubated on ice for 15 min. Following cell lysis, the mixtures were then centrifuged at 4 °C at 10,000 rpm for 10 min. The supernatant was incubated with p-nitrophenyl phosphate (pNpp, Sigma-Aldrich, St Louis, MO, USA) liquid at 1:1 ratio and 37 °C in a 96-well plate for 2 h. The pNpp would be hydrolyzed by ALP to p-nitrophenol (pNp). The addition of 100 µL of 0.2M NaOH was to stop the hydrolysis. The optical density of the pNp was read at 405 nm using the Varioskan^TM^ Multiplate reader. 

### 4.8. Western Blot Analysis

This analysis aided to detect the presence of ALP, early differentiation marker. Sodium dodecyl sulfate-polyacrylamide gel electrophoresis (SDS-PAGE) was conducted at 170 V for 75 min at a loading volume of 30 µL of heat-denatured supernatant (90 °C for 10 min) from the aforementioned cell lysis reaction (PRO-PREP^TM^) using the NuPAGE Novex Bis-Tris Precast gel (Life Technologies, Carlsbad, CA, USA). The size-separated protein sample was then electro-blotted at 25 V for 90 min using the XCell II™ Blot Module (Life Technologies, Carlsbad, CA, USA). The electro-blotted nitrocellulose membrane was then rinsed with deionized water prior to soaking in blocking buffer of 4% (*w*/*v*) milk in phosphate buffered saline-0.05% Tween-20 (PBS-T) solution. After horizontal shaking at 100 rpm for 30 min, mouse primary anti-ALP antibody ab54778 (Abcam, Eugene, OR, USA) was added at 1:2000 dilution to the blocking buffer and horizontally shaken at 100 rpm for 5 min before storing at 4 °C overnight. Subsequently, it was horizontally shaken at 100 rpm for 2 h prior to three 10 mL PBS-T washing steps of 4-min interval each. Thereafter, horseradish peroxidase linked-anti-mouse secondary antibody ab9484 (Abcam, Eugene, OR, USA) was added at 1:5000 dilution in 10 mL of PBS-T solution. The reaction was maintained on the horizontal shaker at 100 rpm for 2 h. This was followed by four 10 mL PBS-T washing steps of 4-min interval each before a final wash with deionized water. The membrane was washed with 1 mL of 3,3′,5,5′-tetramethylbenzidine (TMB) substrate until band was visibly observed. The partially dried membrane was then captured using GS-800 densitometer (Bio-Rad, Hercules, CA, USA). 

### 4.9. Statistical Analysis

MTT and ALP assays were statistically analyzed with two-way ANOVA in Randomized Complete Block Design using the software GenStat 17th Edition (32 bit). 

## 5. Conclusions

Glutaraldehyde crosslinked CS scaffolds dispersed with GO offered promising in vitro bioactivities in terms of MG63 cell adhesion, proliferation, and differentiation. CS-GO2’s bioactivities were found to be more associated with surface nanotopography while those of CS-GO6 were more likely to be contributed by surface physicochemistry bestowed by the oxygen-containing chemical groups of GO. Nonetheless, dispersion of GO in CS solution garners further improvement so that glutaraldehyde crosslinked CS-GO nanocomposites can act synergistically via organized nanotopography coupled with enhanced surface physiochemistry in terms of MG63 cell adhesion, proliferation, and osteogenesis. 

## Figures and Tables

**Figure 1 ijms-21-05202-f001:**
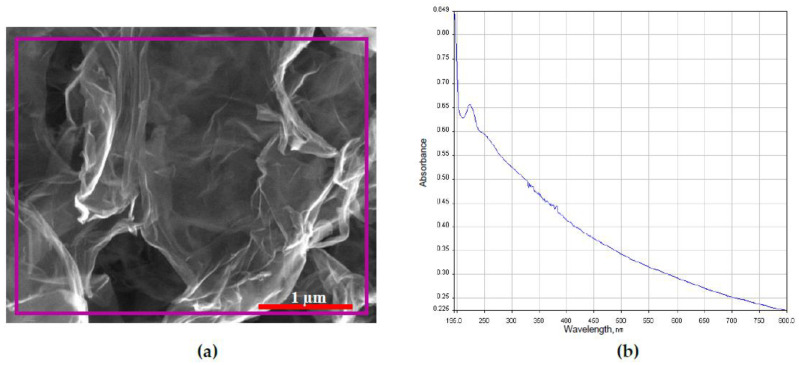
(**a**) FESEM image with boxed area (purple line) of which the EDX analysis performed and (**b**) UV–VIS absorption spectrum of GO.

**Figure 2 ijms-21-05202-f002:**
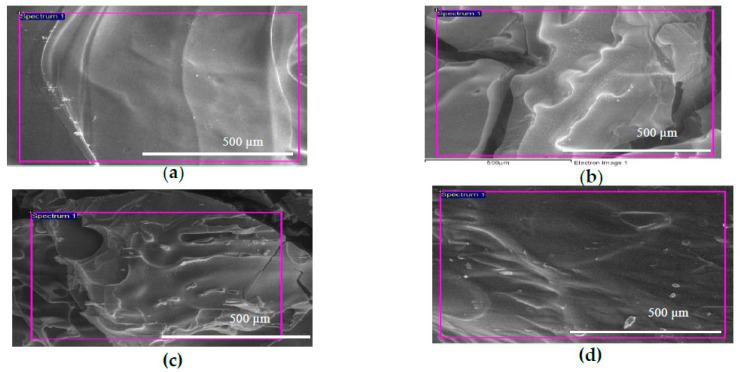
FESEM image of (**a**) pristine GTA crosslinked CS scaffold (**b**–**d**) CS-GO2, CS-GO4, CS-GO6, GTA crosslinked CS scaffolds dispersed with GO at 2, 4, and 6 wt % respectively.

**Figure 3 ijms-21-05202-f003:**
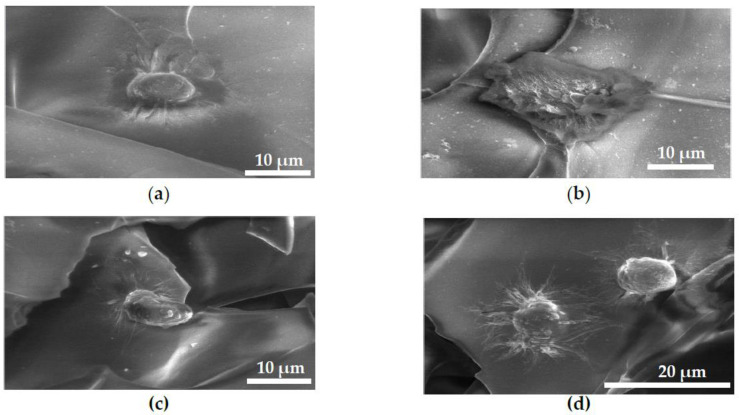
FESEM images of MG63 cells fixed on GTA crosslinked scaffolds, (**a**–**d**) CS, CS-GO2, CS-GO4 and CS-GO6, respectively.

**Figure 4 ijms-21-05202-f004:**
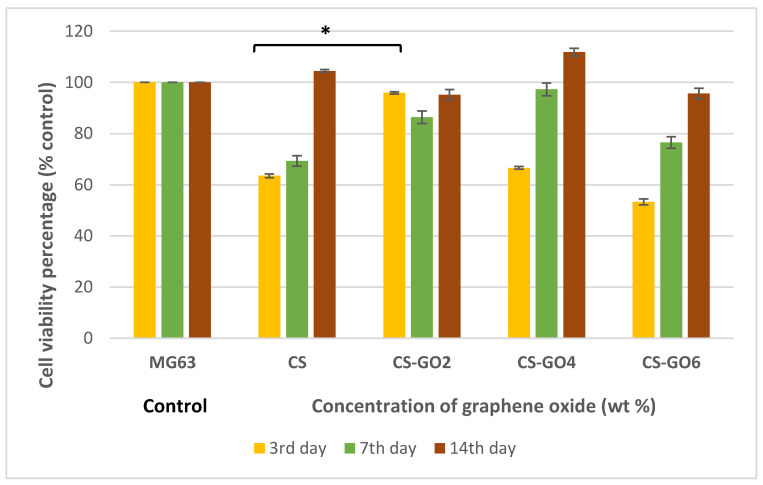
Histogram showing MG63 cell viability percentages (%) against control on 3rd, 7th, and 14th day post seeding on GTA crosslinked CS-GO scaffolds. Error bars represent standard error of means (SEM). Independent experiments were conducted in triplicate (* *p* < 0.001).

**Figure 5 ijms-21-05202-f005:**
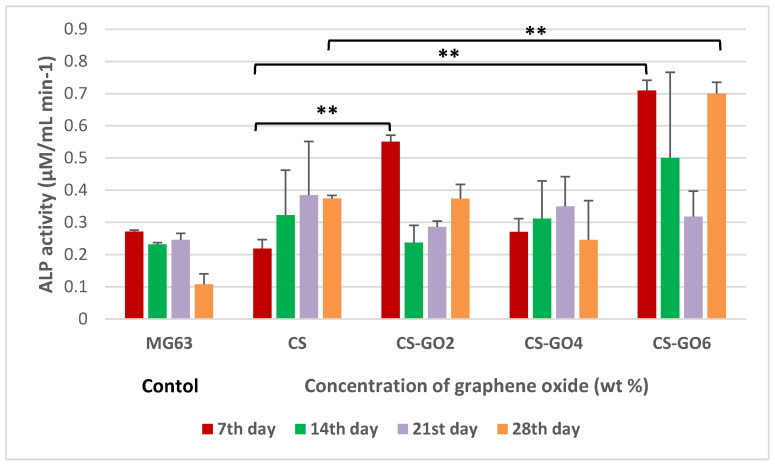
Histogram showing alkaline phosphatase (ALP) activity of MG63 cells on scaffolds on 7th, 14th, 21st, and 28th day post-seeding. Error bars represent standard error of means (SEM). Independent experiments were conducted in triplicate (** *p* < 0.001).

**Figure 6 ijms-21-05202-f006:**
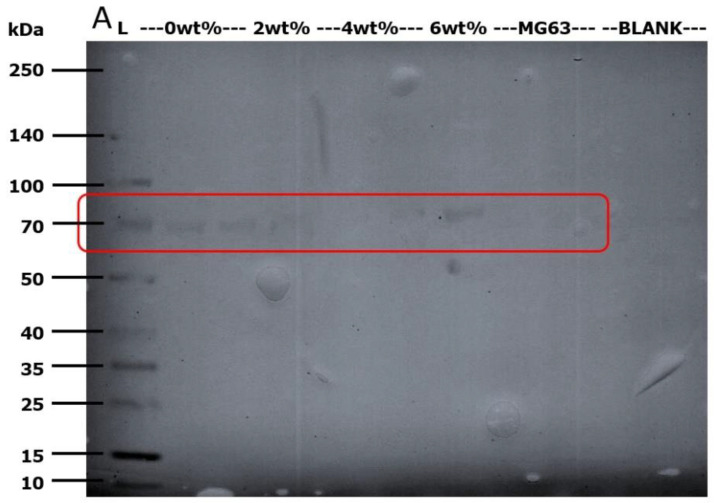
Nitrocellulose membrane from western immunodetection on 7th day post-incubation protein extract (A). The lanes starting from left indicate (L): the Multicolor Broad Range Protein Ladder (Fermentas). CS—2 lanes, CS-GO2—1 lane, CS-GO4—2 lanes, CS-GO6—1 lane, MG63 (control)—1 lanes and BLANK—2 lanes.

**Table 1 ijms-21-05202-t001:** Elemental composition of GO

Element	Atomic %
Carbon	72.80
Oxygen	27.09
Sulfur	0.11

**Table 2 ijms-21-05202-t002:** GTA crosslinked CS scaffolds with different wt % of GO

Scaffold	Wt % of GO
CS	0
CS-GO2	2
CS-GO4	4
CS-GO6	6

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
