# Peer review of "Preliminary In Vitro Evaluation of Chitosan–Graphene Oxide Scaffolds on Osteoblastic Adhesion, Proliferation, and Early Differentiation"

_ijms, 2020, doi:10.3390/ijms21155202_

Round 1

Reviewer 1 Report

The submitted manuscript presents the fabrication and characterization of chitosan-graphene oxide composites to assess MG63 cells response. The proposal is not novel considering both the selected materials and the manufacturing technique. In addition, the collected results can unlikely provide a significant advancement in the field, thus implying a limited further coverage of the study. Furthermore, the GO content is very high, but the Authors did not state the rationale for this experimental condition.

In the preparation stage GO was directly blended to CS solution and nothing is reported about a possible preliminary disagglomeration procedure for the nanofiller, e.g. ultrasonic bath: this issue should be addressed. In addition, it was also reported that scaffolds were rehydrated with different ethanol concentrations for different time periods: it is not clear the rationale for the combinations here considered and, again, nothing about it is reported.

In Par. 4.3 it was stated “The scaffolds with different weight percent of GO were sterilized with ultraviolet ray treatment”, while in Par. 4.2 “A final step of sterilizing each crosslinked scaffold in 1 mL of 70% (v/v) ethanol for 30 min was conducted…”: this should be clarified.

The formula reported in Par. 4.6 is not correct.

In Par. 2.1 the EDX results were reported with respect to carbon and oxygen, but it is not clear the actual added value of this analysis considering their abundance in chitosan. Most interestingly, the Authors should explain why GO is not observed on CS-GO4 and CS-GO6 surfaces when it is more abundant in those samples.

Fig. 2 shows MG63 cells on the scaffolds, but they are very isolated and this does not seem a valuable result. SEM micrographs at lower magnifications could contribute to properly evaluate the actual cell response to the proposed scaffolds.

In Par. 4.8 it was reported that the statistical analysis was carried out by means of two-way ANOVA, while in Par. 2.3 also the one-way ANOVA was used. Fig. 3 is not easily readable, as the related text firstly reported a comparison between CS-GO4 on the 14th day and CS-GO6 on the 3rd day, then CS-GO2 was compared with respect to all scaffolds. The Authors should more homogenously compare the collected data.

The sentences “A general observation was that ALP activities of MG63 cells on all scaffolds on each day post-incubation were low, at the nano molar levels instead of micro molar. However, based on the findings of this study, GO loading did give a certain level of enhancement of the osteogenic properties of GTA crosslinked CS scaffolds” should be rephrased to improve their readability.

Fig. 5 should be enhanced because, due its low quality, it is not possible to agree with the Authors.

Author Response

Comment 1: The submitted manuscript presents the fabrication and characterization of chitosan-graphene oxide composites to assess MG63 cells response. The proposal is not novel considering both the selected materials and the manufacturing technique. In addition, the collected results can unlikely provide a significant advancement in the field, thus implying a limited further coverage of the study. Furthermore, the GO content is very high, but the Authors did not state the rationale for this experimental condition.

Response:Thank you so much for your comment. The rationale of this study is to reinforce chitosan-based scaffolds with GO for enhanced in vitro biocompatibility. The following statement has been added in the Introduction. “This study aimed to reinforce CS scaffolds with GO at weight percent of 2, 4 and 6 which were further crosslinked with glutaraldehyde (GTA)…. Like alkaline phosphatase (ALP) [23-24].”

The highest GO content (6 wt%) in this study was justified by reference [22].

Aidun, A.; Firoozabady, A. S.; Moharrami, M.; Ahmadi, A.; Haghihipour, N.; Bonakdar, S.; Faghihi, S. Graphene oxide incorporated polycaprolactone/chitosan/collagen electrospun scaffold: Enhanced osteogenic properties for bone tissue engineering. Artif. 2019, 43, E264-E281.

Comment 2: In the preparation stage GO was directly blended to CS solution and nothing is reported about a possible preliminary disagglomeration procedure for the nanofiller, e.g. ultrasonic bath: this issue should be addressed. In addition, it was also reported that scaffolds were rehydrated with different ethanol concentrations for different time periods: it is not clear the rationale for the combinations here considered and, again, nothing about it is reported.

Response: GO was first dispersed in deionized water using an ultrasonic bath before the blending with chitosan solution. The following statement has been added in section 4.2. “Prior to blending with CS, different wt% of GO were suspended in deionized water for 5 min using ultrasonic bath (Elmasonic S180H, Germany).”

The rehydration of the scaffolds served the purpose of sterilizing and neutralizing the CS scaffolds, as CS was previously dissolved in diluted acetic acid. Besides, it could help the scaffolds to regain their structure after their absorption of water from 70% and 50% alcohols. The rationale for the alcohol combinations can be justified by reference [41].

Madihally, S.; Matthew, H.W.T. Porous chitosan scaffolds for tissue engineering. Biomater. 1999, 20, 1133-1142. The following statement has been added in section 4.2.

“The rehydration of CS scaffolds using alcohols allowed direct sterilization of scaffolds and hydrated CS scaffolds were reported to be spongy and very flexible.”

Comment 3: In Par. 4.3 it was stated “The scaffolds with different weight percent of GO were sterilized with ultraviolet ray treatment”, while in Par. 4.2 “A final step of sterilizing each crosslinked scaffold in 1 mL of 70% (v/v) ethanol for 30 min was conducted…”: this should be clarified.

Response: In Par. 4.3, the sentence “The scaffolds with different weight percent of GO were sterilized with ultraviolet ray treatment” has been removed for clarity. Please refer to Par. 4.2 for scaffold sterilizing step.

Comment 4: The formula reported in Par. 4.6 is not correct.

Response: This formula has been removed from the revised manuscript.

Comment 5: In Par. 2.1 the EDX results were reported with respect to carbon and oxygen, but it is not clear the actual added value of this analysis considering their abundance in chitosan. Most interestingly, the Authors should explain why GO is not observed on CS-GO4 and CS-GO6 surfaces when it is more abundant in those samples.

Response: The following explanation has been included in discussion part to explain the EDX results in Par 2.1. “It is suggested that GO dispersion is more even in CS solution containing 2 wt% of GO. When the wt% of GO is increased to 4 and 6, the agglomeration of GO becomes dominant due to high surface energy. The dispersive force by magnetic stir bar is not strong enough to prevent the agglomeration. Instead of being well dispersed in the solution, GO stay as clumps and deposit at other parts of CS-GO scaffolds. As a result, GO is not detected on the surface of CS-GO4 and CS-GO6.”

Comment 6: Fig. 2 shows MG63 cells on the scaffolds, but they are very isolated and this does not seem a valuable result. SEM micrographs at lower magnifications could contribute to properly evaluate the actual cell response to the proposed scaffolds.

Response: At higher magnification, the filopodia extension of the cells cultured on CS, GO4 and GO6 scaffolds can be clearly observed. With cellular filopodia extension, cells on these scaffolds demonstrated certain degree of migration which better explained the cytocompatibility of these scaffolds. These morphologies would not be clearly detected at lower magnification.

Comment 7: In Par. 4.8 it was reported that the statistical analysis was carried out by means of two-way ANOVA, while in Par. 2.3 also the one-way ANOVA was used. Fig. 3 is not easily readable, as the related text firstly reported a comparison between CS-GO4 on the 14th day and CS-GO6 on the 3rd day, then CS-GO2 was compared with respect to all scaffolds. The Authors should more homogenously compare the collected data.

Response: After addition of Par 2.1 for the characterization of GO, the subsequent headings have been revised. Par 2.3 has been changed to Par 2.4. The one-way ANOVA in Par 2.4 (MTT Assay) has been removed from the revised manuscript, as the statistical analysis carried out in this study was two-way ANOVA. Besides, the comparison between CS-GO4 on the 14th day and CS-GO6 on the 3rd day has been removed.

Comment 8: The sentences “A general observation was that ALP activities of MG63 cells on all scaffolds on each day post-incubation were low, at the nano molar levels instead of micro molar. However, based on the findings of this study, GO loading did give a certain level of enhancement of the osteogenic properties of GTA crosslinked CS scaffolds” should be rephrased to improve their readability.

Response:

The sentences have been rephrased to “ALP activities of MG63 cells on all scaffolds were observed on each day post-incubation. However, as shown in Figure 5, GO loading enhanced the osteogenic properties of GTA crosslinked CS scaffolds in terms of ALP activities.” Figure 4 has been changed to Figure 5.

Comment 9: Fig. 5 should be enhanced because, due its low quality, it is not possible to agree with the Authors.

Response: An image with higher resolution has been added in the revised manuscript. Clear bands were found in Figure 6 which was previously Figure 5.

Reviewer 2 Report

Authors claim the incorporation of graphene oxide into chitosan matrix to increase the cell adhesion, proliferation and early differentiation of MG63 human osteosarcoma cell line  respectively.

Results showed that glutaraldehyde crosslinked chitosan scaffolds loaded with graphene at 2 weight percent mediated promising initial MG63 cell adhesion, proliferation, and early differentiation on as early as 7th day post-seeding.  

Concluded that chitosan scaffolds loaded with graphene oxide at 2 weight percent is considered worthwhile to be developed into functional bone graft substitute in the future.

Some input may up grade this manuscript.

  1. Generally, collagen ECM, GAG and BCP are most concerned as the bone repair materials. Authors are requested to evaluate and compared each other.
  2. GTA is toxic to cells. Authors may have to figure out it.
  3. Mechanisms of CS-graphene induced bone formation needs to address in detail. Especially in discussion section. 

Author Response

Comment 1: Generally, collagen ECM, GAG and BCP are most concerned as the bone repair materials. Authors are requested to evaluate and compared each other.

Response: Thank you so much for your comment. As suggested by the title of the manuscript, this study aimed to evaluate the preliminary in vitro efficacy of the GO scaffolds. Therefore, collagen ECM, GAG and BCP were not evaluated in this study.

Comment 2: GTA is toxic to cells. Authors may have to figure out it.

Response: Authors are aware of the cytotoxicity of GTA. However, the concentration of GTA attempted in this study was low. All CS-GO scaffolds were rinsed with excessive deionized water to remove GTA after crosslinking. Most importantly, MG63 cells were able to grow on CS-GO scaffolds by showing cell viability of more than 70% after 7 and 14 days of incubation. This demonstrated that MG63 cells could handle the chemical stress induced by GTA if there was any GTA in the scaffolds.  

Comment 3: Mechanisms of CS-graphene induced bone formation needs to address in detail. Especially in discussion section. 

Response: The mechanism of CS-graphene scaffolds in terms of inducing bone formation has been further elaborated in the discussion section. The following explanation has been included in the revised manuscript: “The mechanism behind the enhancement of osteogenesis….. ALP activity and mineralization in mouse osteoblasts [36-37].”

Reviewer 3 Report

This manuscript presents an experimental study regarding the preparation of 3D glutaraldehyde crosslinked chitosan with insertion with graphene nanoparticles. Scaffolds are studied in terms of morphology and composition and regarding their interaction with human osteosarcoma cell line MG63. The presented data are of interest for the scientific community involved in chitosan and nanoparticles applications as bone tissue regeneration materials. 

Some clarifications are necessary:

  1. In Section 3 Discussions

The authors explain the missing of the interconnected porous structure which should have been formed via the freeze-drying processes by the freeze-drying conditions (chitosan concentration). I think that the porosity collapsing is the result of the drying process after crosslinking (subsequently desiccated with silica gels until complete dryness). The authors should explain why they have chosen this method and not a new freeze-drying process.

  1. In Section 4. Materials and Methods

The grapheme oxide characteristics should be included.

  1. In Section 4. Materials and Methods

The authors should indicate if the crosslinked scaffolds was purified with deionized water by a single immersions or if the deionized water was changed at different periods of time

Author Response

Comment 1: The authors explain the missing of the interconnected porous structure which should have been formed via the freeze-drying processes by the freeze-drying conditions (chitosan concentration). I think that the porosity collapsing is the result of the drying process after crosslinking (subsequently desiccated with silica gels until complete dryness). The authors should explain why they have chosen this method and not a new freeze-drying process.

Response: Authors dried CS scaffolds with TiO2 nanotubes as the inorganic filler using silica gels in previous study. The porous structure of CS-TiO2 nanotubes scaffold was still preserved and reported in reference [29].

Lim, S.S.; Chai, C.Y.; Loh, H-S. In vitro evaluation of osteoblast adhesion, proliferation and differentiation on chitosan-TiO2 nanotubes scaffolds with Ca2+ ions. Mater Sci Eng: C. 2017, 76, 144-152.

The weight percent of TiO2 nanotubes in the CS scaffolds was 16 wt% which contributed to the structure integrity of the scaffolds. In addition to the chitosan concentration, the porosity collapsing in CS-GO scaffolds was also resulted from the low GO content in the scaffolds. The following statements have been added in the discussion part. “Besides, the weight percent of GO is not high enough to retain the porous structure of CS scaffolds. CS scaffolds with TiO2 nanotubes as filler in a previous study showed porous structure [29]. The weight percent of TiO2 nanotubes in the scaffolds was as high as 16 wt%. However, the weight percent of GO has to be kept not more than 6 in this study to ensure the cytocompatibility of CS scaffolds.”  

Comment 2: The grapheme oxide characteristics should be included.

Response: Graphene oxide was characterized by using FESEM, EDX and UV/Vis spectrophotometer. The characterization results have been added in section 4.1.

Comment 3: The authors should indicate if the crosslinked scaffolds was purified with deionized water by a single immersions or if the deionized water was changed at different periods of time.

Response:The sentence has been rephrased as follows:

“Each of the crosslinked scaffolds was then rinsed with three washes of 200-300 mL of deionized water and subsequently desiccated with silica gels until complete dryness.”

Reviewer 4 Report

The manuscript describes the production of several chitosan-based scaffolds which incorporate graphene oxide. MG63 cells are then grown on these scaffolds in order to assess GO impact on cell adhesion, proliferation and osteoinduction.

This manuscript has several serious problems which are highlighted below:

1. As the authors indicate (p 3 line 1O6), it is not possible to see any evidence of GO particles on SEM images for CS-GO4 and CS-GO6. They do however suggest that the SEM images show that there is a difference in surface roughness for scaffolds incorporating GO. This is not something that can really be quantified from 2D SEM images, and a more appropriate technique for assessing and quantifying roughness should be considered. Looking at the SEM images the surface appearance for CS-GO6 doesn’t seem different to CS alone (Figure 1 a and d).

EDX also shows no difference between control and CS-GO4 and CS-GO6 (Table 1), and cell morphology is rounded on these two scaffolds as well in a similar manner to the control (Figure 2 a, c and d).

Increase of metabolic activity of cells on CS-GO4 is only significant when compared to CS-GO6 (rather than CS alone) (Figure 3). It is not valid to compare CS-GO4 and CS-GO6 in this way, and all comparisons should be made to the CS control. When this is done there is no difference between CS-GO4, CS-GO6 and control.

All the above together suggests that there is no evidence of any GO particles present in CS-GO4 and CS-GO6 scaffolds meaning the discussion of these results and suggestion that inclusion of GO in these scaffolds influences cell growth is unsubstantiated. The only scaffold where GO inclusion may have occurred seems to be CS-GO2 and only these results can be considered.

Statements that describe GO presence in scaffolds other than CS-GO2 should be removed or better substantiated.

2. The authors also discuss the homogeneity of GO dispersion in CS-GO2 and compare it to that for CS-GO4 and CS-GO6. They suggest that there is an uneven distribution of particles for the CS-GO4 and CS-GO6 and it is more homogenous for CS-GO2. Given that there seems to be no evidence for any inclusion of particles at all on the O4 and O6 this claim appears unfounded and should be removed.

3. The authors speculate that incorporation of GO affects the nanotopography of the scaffold (page 7 line 225 onwards). This appears to be based on the belief that the GO on the CS-GO2 is arranged in a more orderly fashion (page 7 line 226). Figure 1b is apparently demonstrating this however I can see no indication of an orderly arrangement in this figure. If the authors have clearer images which show this, they should include them here, otherwise it does not seem reasonable to make this comment.

4. The authors suggest that GO incorporation increases “polarity and wettability” (page 7 line 216) and that this improves cell adhesion. A simple way to measure this would be to test the contact angle of water on the different surfaces. This, or another similar assessment would provide more information on the impact of GO incorporation speculated in this paper.

5. The materials and methods section seems to suggest that the ALP standard curve may have been performed at µM concentrations (page 10 line 345) and the results extrapolated from this standard curve. If so, this would mean that the reported pM values and the conclusions drawn from this are unreliable, and differences between samples cannot be accurately identified. It is necessary to show that the standard curve covers the appropriate range for the values reported and to demonstrate that the values reported are within the range that can be detected by this assay.

6. As the values of p-nitrophenol produced were so low (less than 1 nM) when it would be expected to be several orders of magnitude greater if osteogenesis had occurred, it does not seem reasonable to suggest that any osteogenesis has taken place or that there has been an enhancement of ALP activity by any of the scaffolds. The statements that ascribe changes in ALP (page 5 lines 153 to 162) to osteoinduction cannot be made with these levels of ALP

This lack of osteogenic induction also seems to be mirrored by the Western blot ALP result which shows only the faintest indication of a band of the appropriate size. It cannot therefore be said from the results reported that the cells on any of the scaffolds have undergone any osteogenic differentiation. The authors need to provide stronger evidence of osteogenic induction or change their analysis of the data to reflect that there is no evidence that this has occurred. Can osteoinduction be verified in any other way – for example by assessing mineralisation via calcium deposition (Alizarin red)?

7. The only MTT result which showed an increase in metabolic activity compared to the control was for CS-G02 at day 3, after that there is no difference between any of the scaffolds and the control (Figure 3). The authors then suggest that further proliferation decreases compared to the control because differentiation is then taking place, however since this does not seem to be confirmed by the ALP results – see point 6, this again needs further evidence to substantiate the claim.

Furthermore, MTT does not measure cell survival (page 4 lines 127, 129, 131, Figure 3 and legend, page 5 line 155, page 7 line 237, page 8 line 261) but does measure cell metabolic activity. After 3 days or more a change in metabolic activity compared to control could be due to increased cellular adhesion, increased cell proliferation or a change in metabolic activity within the population of cells. It cannot however indicate the percentage of cells that have survived and this needs to be clarified throughout the document. 

There are also several minor points:

8. Issues with English clarity -the writing can be quite long-winded and verbose, errors in grammar and incorrect word usage sometimes arise as a result.  The manuscript would be significantly clearer if this verbosity was reduced

e.g. P. 1 Introduction – First sentence doesn’t really make sense – “Tissue engineering was a jargon invented when scientists have, throughout the years, successfully discovered the solution for the misery underlying the limitations of the autografts and allografts approach in regenerating bone defect areas, either arise due to congenital anomalies or induced traumatically”

Other errors

p. 5 line 153 Interestingly, despite at a statistically insignificant - delete “at” p. 5 line 160 before ascending to the a level of ALP – delete “a”

9. Scale bar on Fig 1 is too small to read

10. Remove words “Wu et al, 2014” from p. 6 line 174

Author Response

Comment 1(a): As the authors indicate (p 3 line 1O6), it is not possible to see any evidence of GO particles on SEM images for CS-GO4 and CS-GO6. They do however suggest that the SEM images show that there is a difference in surface roughness for scaffolds incorporating GO. This is not something that can really be quantified from 2D SEM images, and a more appropriate technique for assessing and quantifying roughness should be considered. Looking at the SEM images the surface appearance for CS-GO6 doesn’t seem different to CS alone (Figure 1 a and d). EDX also shows no difference between control and CS-GO4 and CS-GO6 (Table 1), and cell morphology is rounded on these two scaffolds as well in a similar manner to the control (Figure 2 a, c and d).

Response:The following statements have been added in discussion to explain the nonexistence of GO on the surface of CS-GO4 and CS-GO6 scaffolds. “It is suggested that GO dispersion is more even in CS solution containing 2 wt% of GO. When the wt% of GO is increased to 4 and 6, the agglomeration of GO becomes dominant due to high surface energy. The dispersive force by magnetic stir bar is not strong enough to prevent the agglomeration. Instead of being well dispersed in the solution, GO stay as clumps and deposit at other parts of CS-GO scaffolds. As a result, GO is not detected on the surface of CS-GO4 and CS-GO6.”

Comment 1(b): Increase of metabolic activity of cells on CS-GO4 is only significant when compared to CS-GO6 (rather than CS alone) (Figure 3). It is not valid to compare CS-GO4 and CS-GO6 in this way, and all comparisons should be made to the CS control. When this is done there is no difference between CS-GO4, CS-GO6 and control.

Response: Thank you so much for the suggestion. All comparisons in term of metabolic activity of cells have been made to the CS control in the revised manuscript.

Comment 1(c): All the above together suggests that there is no evidence of any GO particles present in CS-GO4 and CS-GO6 scaffolds meaning the discussion of these results and suggestion that inclusion of GO in these scaffolds influences cell growth is unsubstantiated. The only scaffold where GO inclusion may have occurred seems to be CS-GO2 and only these results can be considered. Statements that describe GO presence in scaffolds other than CS-GO2 should be removed or better substantiated.

Response:

The following statement has been included in discussion to address this comment. “GO was not detected on the surface of CS-GO4 and CS-GO6 scaffolds due to the agglomeration of GO. However, this does not mean GO is not in CS-GO4 and CS-GO6 scaffolds. The release of GO clumps in both scaffolds starts upon the degradation of chitosan matrix. Even the clumps are not released, they remain in the chitosan matrix facilitating sites for osteoblastic adhesion which will be followed by proliferation and differentiation.”

Comment 2: The authors also discuss the homogeneity of GO dispersion in CS-GO2 and compare it to that for CS-GO4 and CS-GO6. They suggest that there is an uneven distribution of particles for the CS-GO4 and CS-GO6 and it is more homogenous for CS-GO2. Given that there seems to be no evidence for any inclusion of particles at all on the O4 and O6 this claim appears unfounded and should be removed.

Response: Thank you so much for your comment. Any sentence related to homogeneity of GO in CS scaffolds has been removed from the revised manuscript.

Comment 3: The authors speculate that incorporation of GO affects the nanotopography of the scaffold (page 7 line 225 onwards). This appears to be based on the belief that the GO on the CS-GO2 is arranged in a more orderly fashion (page 7 line 226). Figure 1b is apparently demonstrating this however I can see no indication of an orderly arrangement in this figure. If the authors have clearer images which show this, they should include them here, otherwise it does not seem reasonable to make this comment.

Response: All statements related to nanotopography of the scaffold were removed.

Comment 4: The authors suggest that GO incorporation increases “polarity and wettability” (page 7 line 216) and that this improves cell adhesion. A simple way to measure this would be to test the contact angle of water on the different surfaces. This, or another similar assessment would provide more information on the impact of GO incorporation speculated in this paper.

Response: The polarity of GO in other studies has been widely reported. References [30-31] have been included to prove the polarity of GO in CS scaffolds in this work.

Bacakova, L.; Grausova, L.; Vacik, J.; Kromaka, A.; Biederman, H.; Choukourov, A.; Stary, V. Nanocomposite and Nanostructured Carbon-based Films as Growth Substrates for Bone Cells. Adv Div Ind App Nanocomp. 2011, 2011, 371-408.

Depan, D.; Girase, B.; Shah, J.S.; Misra, R.D.K. Structure-process-property relationship of the polar graphene oxide-mediated cellular response and stimulated growth of osteoblasts on hybrid chitosan network structure nanocomposite scaffolds. Acta Biomater. 2011, 7, 3432-3445.

Comment 5: The materials and methods section seems to suggest that the ALP standard curve may have been performed at µM concentrations (page 10 line 345) and the results extrapolated from this standard curve. If so, this would mean that the reported pM values and the conclusions drawn from this are unreliable, and differences between samples cannot be accurately identified. It is necessary to show that the standard curve covers the appropriate range for the values reported and to demonstrate that the values reported are within the range that can be detected by this assay.

Response: There was an error in the calculation for ALP activities in previous manuscript due to wrong unit conversion. The amount of p-nitrophenol produced was in the range of micromole. The ALP activities of MG63 cultured on CS and CS-GO scaffolds in Figure 5 have been revised.

Comment 6 (a): As the values of p-nitrophenol produced were so low (less than 1 nM) when it would be expected to be several orders of magnitude greater if osteogenesis had occurred, it does not seem reasonable to suggest that any osteogenesis has taken place or that there has been an enhancement of ALP activity by any of the scaffolds. The statements that ascribe changes in ALP (page 5 lines 153 to 162) to osteoinduction cannot be made with these levels of ALP.

Response: The amount of p-nitrophenol produced was in the range of micromole. The ALP activities of MG63 in CS-GO scaffolds show osteogenic induction. This has been revised in the manuscript.

Comment 6(b): This lack of osteogenic induction also seems to be mirrored by the Western blot ALP result which shows only the faintest indication of a band of the appropriate size. It cannot therefore be said from the results reported that the cells on any of the scaffolds have undergone any osteogenic differentiation. The authors need to provide stronger evidence of osteogenic induction or change their analysis of the data to reflect that there is no evidence that this has occurred. Can osteoinduction be verified in any other way – for example by assessing mineralisation via calcium deposition (Alizarin red)?

Response: A western blot image with clearer bands at 80kDa has been added in the revised manuscript. Those bands have proven the osteogenic induction caused by the CS-GO scaffolds.

Comment 7 (a): The only MTT result which showed an increase in metabolic activity compared to the control was for CS-G02 at day 3, after that there is no difference between any of the scaffolds and the control (Figure 3). The authors then suggest that further proliferation decreases compared to the control because differentiation is then taking place, however since this does not seem to be confirmed by the ALP results – see point 6, this again needs further evidence to substantiate the claim.

Response: The claim about the decrease in proliferation of MG63 in CS-GO scaffolds caused by differentiation has been removed from the revised manuscript.

Comment 7(b): Furthermore, MTT does not measure cell survival (page 4 lines 127, 129, 131, Figure 3 and legend, page 5 line 155, page 7 line 237, page 8 line 261) but does measure cell metabolic activity. After 3 days or more a change in metabolic activity compared to control could be due to increased cellular adhesion, increased cell proliferation or a change in metabolic activity within the population of cells. It cannot however indicate the percentage of cells that have survived and this needs to be clarified throughout the document. 

Response: Thank you so much for your explanation. The term “cell survival” has been changed to “cell viability” throughout the revised manuscript.

Comment 8: Issues with English clarity -the writing can be quite long-winded and verbose, errors in grammar and incorrect word usage sometimes arise as a result.  The manuscript would be significantly clearer if this verbosity was reduced.

e.g. P. 1 Introduction – First sentence doesn’t really make sense – “Tissue engineering was a jargon invented when scientists have, throughout the years, successfully discovered the solution for the misery underlying the limitations of the autografts and allografts approach in regenerating bone defect areas, either arise due to congenital anomalies or induced traumatically”.

Response: Thank you so much for your comments. We have improved the sentence structure, grammar and overall quality of the revised manuscript. Some irrelevant write ups have been removed.

The sentence has been rephrased as follows: “Tissue engineering was a novel solution to the misery underlying the limitations of the autografts and allografts approach in regenerating bone defect areas, either arising due to congenital anomalies or induced traumatically”.

Comment 9: p. 5 line 153 Interestingly, despite at a statistically insignificant - delete “at” p. 5 line 160 before ascending to the a level of ALP – delete “a”

Response: Amendments have been made in the stated instances. 

Comment 10: Scale bar on Fig 1 is too small to read.

Response: Bigger scale bar has been inserted into Figure 1.

Comment 11: Remove words “Wu et al, 2014” from p. 6 line 174.

Response: The words have been removed.

Round 2

Reviewer 1 Report

The Authors revised the manuscript, even if a more critical approach was expected.

For instance, regarding the GO amount it was stated that this choice was referred to an already published paper and this does not seem a scientific rationale.

The EDX analysis still does not provide any added value and the response regarding the absence of GO on scaffolds surface is simply not shareble (CS-GO4 and CS-GO6).

Author Response

For instance, regarding the GO amount it was stated that this choice was referred to an already published paper and this does not seem a scientific rationale.

Answer:

Thank you so much for your comment. The rationale of this study was to reinforce CS-GO scaffolds with glutaraldehyde to ensure structural integrity of the scaffolds for osteoblastic functions like adhesion, proliferation and early differentiation. The following statements have been added on p2 line 72-75.

“Further GTA crosslinking on CS-GO scaffolds would be expected to show structural integrity to facilitate osteoblastic functions like adhesion, proliferation and early differentiation. However, the preliminary in vitro efficacy of GTA crosslinked CS-GO scaffolds has not been proven which would be verified in this study”

The EDX analysis still does not provide any added value and the response regarding the absence of GO on scaffolds surface is simply not shareble (CS-GO4 and CS-GO6).

Answer:

The EDX results and any discussion and speculation related to EDX finding have removed from the manuscript. The removed sentences are shown below:

“As a result, GO is not detected on the surface of CS-GO4 and CS-GO6. However, this does not mean GO is not in CS GO4 and CS GO6 scaffolds”

Besides, the following statement has been added in the revised manuscript:

“However, more investigation is needed to confirm the presence of GO particles in other parts of scaffolds.” (p7 line 189-190)

Reviewer 4 Report

The manuscript describes the production of several chitosan-based scaffolds which incorporate graphene oxide. MG63 cells are then grown on these scaffolds in order to assess GO impact on cell adhesion, proliferation and osteoinduction.

This revised manuscript has problems which are highlighted below

  1. Figure 4 In the text (p. 5 line 126) the cellular response to GO particles has been compared to the MG63 control which is presumably these cells growing on tissue culture plastic, and shows no significant difference. However in the graph, significance has been indicated for day 3 CS-GO2, through comparison to CS alone. This is not explained anywhere and this needs to be clarified in both the text and figure legend.
  2. Figure 5, it is again unclear what comparison has been made for the significant differences reported – presumably it is against the CS alone, but this needs to be expressly stated, particularly when the text is comparing to the control (p, 6 line 144).

On discussion of this figure (p. 6 line 135) there is the statement that GO loading enhances osteogenic properties of scaffolds. This statement is too strong and needs to be downgraded to a suggestion that it may have an effect but more work is required to confirm this, since no improvements were seen for CS-GO4; CS-GO2 shows an early response but this improvement is not maintained and GS-O6 has shown an unusual profile of ALP activity which the authors do not attempt to explain.

The authors claim (p. 6 line 145) that ALP activity for CS-GO6 starts off high then plummets across day 14 and 21. However, the large error bars on day 14 mean that actually there is no significant change between days 7 and 14. The only drop that can be reported would be for day 21. Since this is an anomalous result compared to all others the authors should attempt to explain this finding.

  1. The authors speculate the reasons why GO particles are present in GS-04 and GS-06 scaffolds but are not detected (p.7 line 174), however they are unable to provide evidence for this speculation. It therefore needs to be stated in the manuscript that the assumption that GO particles are in the scaffolds cannot be proved, and more work is needed to confirm the presence of these particles. The suggestion that GO clumps are present in the chitosan and act to facilitate osteoblast adhesion, proliferation and differentiation can be a suggestion at most and certainly needs more investigation.

Author Response

Q1: Figure 4 In the text (p. 5 line 126) the cellular response to GO particles has been compared to the MG63 control which is presumably these cells growing on tissue culture plastic, and shows no significant difference. However in the graph, significance has been indicated for day 3 CS-GO2, through comparison to CS alone. This is not explained anywhere and this needs to be clarified in both the text and figure legend.

Answer:

Thank you for your comment. A bar has been included in Figure 4 to indicate the comparison between CS-GO2 and CS scaffolds. For clarification, the following statement has also been revised on p.5 line 127-128.

“As presented in Figure 4, CS-GO2 gives promising cell viability percentage of 95.84% on the 3rd day post-seeding and shows significant difference compared to CS scaffolds (*p<0.001).”

Q2: Figure 5, it is again unclear what comparison has been made for the significant differences reported – presumably it is against the CS alone, but this needs to be expressly stated, particularly when the text is comparing to the control (p, 6 line 144).

On discussion of this figure (p. 6 line 135) there is the statement that GO loading enhances osteogenic properties of scaffolds. This statement is too strong and needs to be downgraded to a suggestion that it may have an effect but more work is required to confirm this, since no improvements were seen for CS-GO4; CS-GO2 shows an early response but this improvement is not maintained and GS-O6 has shown an unusual profile of ALP activity which the authors do not attempt to explain.

The authors claim (p. 6 line 145) that ALP activity for CS-GO6 starts off high then plummets across day 14 and 21. However, the large error bars on day 14 mean that actually there is no significant change between days 7 and 14. The only drop that can be reported would be for day 21. Since this is an anomalous result compared to all others the authors should attempt to explain this finding.

Answer:

Thank you so much for your comment. The comparison, in fact, has been made between CS-GO scaffolds and CS scaffolds. Three bars have been included in Figure 5 to indicate the comparison between CS-GO scaffolds and CS scaffolds.

Therefore, the following statement has been revised as follow:

“ …. a significant (*p<0.001) level of ALP activity on CS-GO2 and CS-GO6 in contrast to CS scaffolds.” p6 line 146-147.

We have revised the following statements which can be found on p6 from line 139-140:

“GO loading may have enhanced the osteogenic properties of scaffolds. Therefore, more studies will be required to verify this.”

The following statement has been revised in the manuscript:

“The ALP activity started off at …… which then plummeted across 14th and on 21st day ….. (Figure 5).”p 6 line 149.

The profile of ALP activity observed in CS-GO6 has been explained as follow:

“The decrease in ALP activity observed in CS-GO6 at day 21 may be caused by the presence of a greater number of mature cells as compared to those undergoing differentiation [26]. On day 28, the increased ALP activity of MG63 on CS-GO6 was observed again which implies bone formation. Therefore, more investigation will be required to confirm this speculation.” p6 line 151-155.

Q3: The authors speculate the reasons why GO particles are present in GS-04 and GS-06 scaffolds but are not detected (p.7 line 174), however they are unable to provide evidence for this speculation. It therefore needs to be stated in the manuscript that the assumption that GO particles are in the scaffolds cannot be proved, and more work is needed to confirm the presence of these particles. The suggestion that GO clumps are present in the chitosan and act to facilitate osteoblast adhesion, proliferation and differentiation can be a suggestion at most and certainly needs more investigation.

Answer:

We have removed the following sentence from the revise manuscript:

“However, this does not mean GO is not in CS GO4 and CS GO6 scaffolds” (p7 line 186)

The following statement has been included:

“However, more investigation is needed to confirm the presence of GO particles in other parts of CS scaffolds.” (p7 line 189-190)